# Recovering Exact Support in Federated lasso without Optimization

**Adarsh Barik**                                          *abarik@nus.edu.sg*
*Institute of Data Science*
*National University of Singapore*

**Jean Honorio**                                          *jhonorio@unimelb.edu.au*
*School of Computing and Information Systems*
*The University of Melbourne*

**Reviewed on OpenReview:** *https://openreview.net/forum?id=S5y26tKlf2*

## Abstract

Federated learning provides a framework to address the challenges of distributed computing, data ownership, and privacy over a large number of distributed clients with low computational and communication capabilities. In this paper, we study the problem of learning the exact support of sparse linear regression in the federated learning setup. We provide a simple communication efficient algorithm that only needs one-shot communication with the centralized server to compute the exact support by majority voting. Our method does not require the clients to solve any optimization problem and thus, can be run on devices with low computational capabilities. Our method is naturally robust to the problems of client failure, model poisoning, and straggling clients. We formally prove that our method requires a number of samples per client that is polynomial with respect to the support size, but independent of the dimension of the problem. We require the number of distributed clients to be logarithmic in the dimension of the problem. For certain classes of predictor variables (e.g. mutually independent, correlated Gaussian, etc.), the overall sample complexity matches the optimal sample complexity of the non-federated centralized setting. Furthermore, our method is easy to implement and has an overall polynomial time complexity.

## 1 Introduction

Modern-day edge devices, with their data acquisition and storage ability, have pushed the need of distributed computing beyond the realms of data centers. Devices such as mobile phones, sensor systems in vehicles, wearable technology, and smart homes, within their limited storage and processing capabilities, can constantly collect data and perform simple computations. However, due to data privacy concerns and limitations on network bandwidth and power, it becomes impractical to transmit all the collected data to a centralized server and conduct centralized training.

The nascent field of federated learning (Konečnỳ et al., 2015; 2016; Brendan et al., 2017; Mohri et al., 2019; Li et al., 2020a) tries to address these concerns. As described by Konečnỳ et al. (2016), federated learning is a machine learning setting where the goal is to train a high-quality centralized model with training data distributed over a large number of clients. Unlike the data centers, the clients collect data samples independently but in a non-i.i.d. fashion. The clients may be highly unbalanced, i.e., the number of samples per client may vary significantly. The clients may also have hardware-related constraints. Although the number of clients could be quite large, each client is typically a simple device that has access to a very small number of data samples and can only conduct very basic computations due to limitations on its processing and power capabilities. Furthermore, since battery power is at a premium, the communication between the client and the centralized server acts as a major bottleneck. Due to these constraints, it is common to encounter straggling and faulty clients in the federated learning setting.

In this work, we study the problem of exact support recovery of sparse linear regression in federated learning without solving any optimization problem. Support recovery in sparse models is of great importance in machine learning as it relates to feature selection. In our setting, none of the clients has the access to necessary number of data samples required for exact support recovery or possess computational capabilities to run complex algorithms. Furthermore, we only allow for one-shot communication between the clients and the centralized server, i.e., clients can send information to the centralized server only once. We propose a novel yet simple algorithm for this setting which uses majority voting and show that local clients can collaboratively recover the exact support of the sparse linear regression model with provable theoretical guarantees.

## 1.1 Related work

The problem of support recovery in sparse linear regression has been well studied for the centralized setting in compressive sensing (See e.g., Foucart & Rauhut, 2017 and references therein) and sparse regression (See e.g., Wainwright, 2009b and references therein). In compressive sensing, for independent sub-Gaussian predictors, $\Omega(s \log d)$ samples are necessary for exact support recovery of a $d$-dimensional parameter vector with $s$ non-zero entries. For sparse regression, Wainwright (2009b) provided the same information-theoretic lower bound for correlated Gaussian predictors. To the best of our knowledge, such a bound does not exist for the general case of *correlated sub-Gaussian* predictors. In the federated setting, data is divided across multiple clients. We define the overall sample complexity in the federated setting as the summation of the sample complexity across all clients. Table 1 shows a comparison of the overall sample complexity of our method in the federated setting to that of the tightest bounds available in the centralized setting running lasso.

The federated learning framework has been used in many empirical studies (Konečný et al., 2015; 2016). As it inherently facilitates distributed computing, it lends itself to be used in a vast range of applications which include but are not limited to deep networks (Brendan et al., 2017), neural networks (Yurochkin et al., 2019; Wang et al., 2020), principal component analysis (Grammenos et al., 2020) and fair resource allocation (Li et al., 2020b). There are also empirical studies that analyze adversarial attacks under the federated learning setting (Bhagoji et al., 2019). On the theoretical side, there are several application-based algorithms that provide convergence rate guarantees. For example, He et al. (2018) provide convergence rate guarantees for linear classification and regression models, Smith et al. (2017b;a) provide similar guarantees for lasso and multi-task learning respectively. Mohri et al. (2019) provide Rademacher-based generalization bounds. Our estimation method at the clients looks similar to marginal regression (See Fan et al., 2008; Genovese et al., 2012 and the works that follow). However, compared to them, we focus on exact support recovery in the federated learning setup. Besides, our analysis and theoretical guarantees hold in the non-asymptotic setting. A detailed survey on the challenges and applications of federated learning can be found in McMahan et al. (2021) and Wang et al. (2021).

Table 1: Comparison of our overall sample complexity of support recovery in sparse regression in the federated setting with existing work in the centralized setting. Notation: $s$ is the number of non-zero entries in the regression parameter vector and $d$ is its dimension. The terms which are independent of $s$ and $d$ are not shown in the order notation.

| Predictor type | Bound in the centralized setting | Our bound |
|---|---|---|
| Mutually independent | $\Theta(s \log d)$ (Wainwright, 2009b) | $\Omega(s \log d)$, Theorem 1 |
| Correlated Gaussian | $\Theta(s \log d)$ (Wainwright, 2009b) | $\Omega(s \log d)$, Appendix G |
| Correlated sub-Gaussian | Not Known | $\Omega(s^2 \log s \log d)$, Theorem 2 |

## 1.2 Our contribution

All the works mentioned above are interesting in their own domain. The existing theoretical works provide guarantees for convergence rates (which guarantees a small mean squared error in the training set provided enough iterations) or generalization bounds (which guarantees a small mean squared error in the testing set provided enough samples). However, the final solution may not match exactly with the true parameter vector.

In this work, we provide provable theoretical guarantees for the exact recovery of the support of the true sparse parameter vector of linear regression in federated learning. Support recovery, i.e., correctly detecting the zero and nonzero entries of the parameter vector, is arguably a more challenging task. We show that for some special classes of predictor variables which include mutually independent or correlated Gaussian random variables, we can do exact support recovery with at least $\Omega(\log d)$ clients and only $\Omega(s)$ data samples per client. Notice that in this case, the overall sample complexity is $\Omega(s \log d)$ which matches the optimal sample complexity of the centralized setting. We also provide novel theoretical results for a general class of correlated sub-Gaussian predictors where we show that if the number of clients is at least $\Omega(\log d)$ and each client has access to at least $\Omega(s^2 \log s)$ data samples, then the support can be recovered exactly with high probability. We propose a simple yet effective method for exact support recovery and prove that the method is *correct* and efficient in terms of *time* and *sample complexity*. McMahan et al. (2021) and Wang et al. (2021) provided several key properties which make federated learning preferable over centralized systems. Our method fulfills many of these key properties:

- **No optimization - low computation:** We do not solve any optimization problem at the client level. All the computations are simple and let us use our method in devices with low computational power.
- **One shot communication and privacy:** Our method is communication efficient. We only need one round communication of at most $d-$bits from the client to the centralized server. As communication is kept to a minimum, very little information about the client is passed to the centralized server.
- **Fault tolerance and aversion to model poisoning and straggling:** Our method is naturally robust to client node failure and averse to rogue and straggling clients.

## 2 Notation and Problem Setup

In this section, we collect the notation which we use throughout this paper. We also formally define the support recovery problem for sparse linear regression in federated learning.

Let $w^* \in \mathbb{R}^d$ be a $d-$dimensional parameter with sparsity $s$, i.e., only $s$ out of $d$ entries of $w^*$ are non-zero. We use $[r]$ as a shorthand notation to denote the set $\{1, 2, \cdots, r\}$. Let $S^*$ be the true support set, i.e., $S^* = \{r | w_r^* \neq 0, r \in [d]\}$. We denote the corresponding complementary non-support set as $S_c^* = \{r | w_r^* = 0, r \in [d]\}$. We will assume that $\min_{j \in S*} |w_j^*| > w^l > 0$ and $\|w^*\|_\infty < w^h$. The first condition is the well-known minimum weight condition (Wainwright, 2009b) which ensures that non-zero entries of $w^*$ are not arbitrarily close to zero which will make inference difficult for any method. The second condition can be written in terms of any $\ell_p$-norm where $p \geqslant 1$. We chose $\ell_\infty$-norm to keep our analysis simple and clean. In federated learning, the data is divided across multiple clients. Assume that there are $g$ clients, each with $n_i$ independent samples, for $i \in [g]$. Note that the data distribution across $g$ clients need not be identical. Each client $i \in [g]$ contains each data sample in the format $(X_i, y_i)$ where $X_i \in \mathbb{R}^d$ are the predictor variables and $y_i \in \mathbb{R}$ is the response variable. The data generation process for each client $i \in [g]$ is as follows:

$$y_i = X_i^\intercal w^* + e_i , \tag{1}$$

where $e_i$ is a zero mean sub-Gaussian additive noise with variance proxy $\eta_i^2$, where $\eta_i > 0$. Note that all the clients share the same parameter vector $w^*$. The $j$-th entry of $X_i$ is denoted by $X_{ij}, \forall i \in [g], j \in [d]$. Each entry $X_{ij}$ of $X_i$ is a zero mean sub-Gaussian random variable with variance proxy $\rho_i^2$, where $\rho_i > 0$. We denote covariance matrix for $X_i$ as $\Sigma^i \in \mathbb{R}^{d \times d}$ with diagonal entries $\Sigma_{jj}^i \equiv (\sigma_{jj}^i)^2, \forall j \in [d]$ and non-diagonal entries $\Sigma_{jk}^i \equiv \sigma_{jk}^i, \forall j, k \in [d], j \neq k$. If predictor variables are mutually independent then $\sigma_{jk}^i = 0, \forall i \in [g], j, k \in [d], j \neq k$. The $t$-th sample of the $i$-th client is denoted by $(X_i^t, y_i^t), \forall i \in [g], t \in [n_i]$. We note that $X_i^t \in \mathbb{R}^d$ and $y_i^t \in \mathbb{R}$ and denote $j$-th entry of $X_i^t$ as $X_{ij}^t$. Notice that the data distributions for $(X_i, y_i)$ can vary a lot across the clients by varying $\rho_i$ and $\eta_i$, as well as the specific sub-Gaussian probability distribution. The class of sub-Gaussian variates includes for instance Gaussian variables, any bounded random variable (e.g., Bernoulli, multinomial, uniform), any random variable with strictly log-concave density, and any finite mixture of sub-Gaussian variables. Similarly, data samples can be distributed unevenly across the clients by

varying $n_i$. In subsequent sections, we use $\mathbb{P}(A)$ to denote the probability of the event $A$ and $\mathbb{E}(A)$ to denote the expectation of the random variable $A$. Figure 1a shows our setup and compares it with a centralized server running lasso in Figure 1b. Notice how each client only sends a maximum of $d$ bits to the centralized server in Figure 1a and maintains the confidentiality of locally collected data. This is unlike the centralized setting where the centralized server has access to all the data.

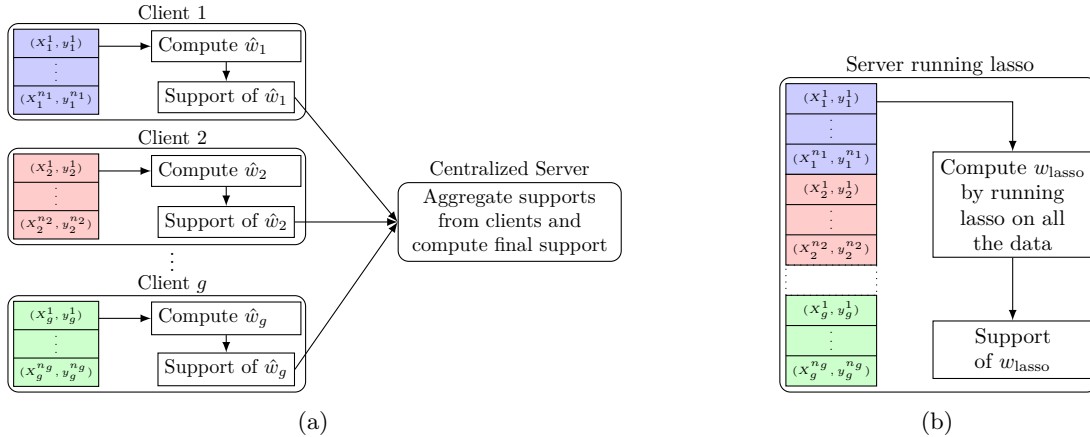

Figure 1: (Left 1a) Support recovery in our federated sparse regression framework. (Right 1b) Support recovery in the centralized sparse regression framework using lasso.

## 3 Problem Statement

For our problem, we assume that each client has access to $n_i = o(s \log d)$ samples, $\forall i \in [g]$. That is, the number of samples $n_i$ grows strictly slower than $s \log d$. Otherwise, the support can be trivially recovered by using compressed sensing methods in the client with $n_i = \mathcal{O}(s \log d)$ which is the order of necessary and sufficient number of samples required for exact support recovery in linear regression setup (Wainwright, 2009a;b). Furthermore, we assume that each of our clients can only do very simple computations and can only do one-shot communication with the centralized server, i.e., each client can only send at most $d$-bits to the centralized server. Considering the above requirements, we are interested in answering the following question:

**Problem Statement 1** (Exact Support Recovery)**.** *Given that each client contains $n_i = o(s \log d)$ data samples generated through the process described in equation (1), is it possible to efficiently recover the true support of the s-sparse shared parameter vector $w^* \in \mathbb{R}^d$ by collecting d-bits of information from every client only once with provable theoretical guarantees.*

The efficiency in exact recovery means that the sample complexity per client should be $o(s \log d)$ and that our algorithm should have polynomial time complexity and should also be easy to implement.

## 4 Our Method

In this section, we present a simple algorithm to solve problem 1. Our main idea is that estimation at the client level can be incorrect for every client but this information can still be aggregated in a careful manner to compute the true support.

### 4.1 Client level computations

Each client tries to estimate the support of $w^*$ using $n_i$ independent samples available to it. As mentioned previously, $n_i, \forall i \in [g]$ is not sufficient to compute correct support of $w^*$ using any method possible (Wainwright, 2009a). Let $\hat{w}_i \in \mathbb{R}^d$ be the estimate of $w^*$ computed by each client $i$. Let $S_i = \{j | \hat{w}_{ij} \neq 0, j \in [d]\}$ be the support of $\hat{w}_i$. Each server communicates the computed support (at most $d$ bits) to a centralized

server which then computes the final support of $w^*$. The centralized server receives $S_i$ from each client and computes the final support $S = f(S_1, S_2, \cdots, S_g)$. Each client $i, \forall i \in [g]$ computes $\hat{w}_i$ in the following way:

$$\forall i \in [g], j \in [d], \ \hat{w}_{ij} = \frac{1}{\hat{\sigma}_{ij}} \text{sign}(\hat{\alpha}_{ij}) \max(0, |\hat{\alpha}_{ij}| - \lambda_{ij}), \tag{2}$$

where $\hat{w}_{ij}$ is $j$-th entry of $\hat{w}_i$ and $\lambda_{ij} > 0$ is a regularization parameter. While it is possible to compute a feasible $\lambda_{ij}$ for each client $i$ and entry $j$, we will present a more practical choice of a single $\lambda_{ij} = \lambda$ across all clients and entries. Moreover, computing $\hat{\sigma}_{ij}$ is not required to estimate the support but we keep all the terms in equation 2 for clarity and completion. We define $\hat{\sigma}_{ij}$ and $\hat{\alpha}_{ij}$ as follows:

$$\hat{\sigma}_{ij} \triangleq \frac{1}{n_i} \sum_{t=1}^{n_i} (X_{ij}^t)^2, \quad \hat{\alpha}_{ij} \triangleq \frac{1}{n_i} \sum_{t=1}^{n_i} y_i^t X_{ij}^t \tag{3}$$

These are simple calculations and can be done in $\mathcal{O}(dn_i)$ run time at each client. If $n_i$ can be kept small (which we will show later), this can be done even by a device with low computational ability. The choice of this exact form of $\hat{w}_{ij}$ in equation (2) is not arbitrary. To get the intuition behind our choice, consider the following $\ell_1$-regularized (sparse) linear regression problem at each client.

$$(\forall i \in [g]), \ \hat{w}_i = \arg \min_w \frac{1}{n_i} \sum_{t=1}^{n_i} (w^\intercal X_i^t - y_i^t)^2 + \|\Lambda_i \odot w\|_1, \tag{4}$$

where $\| \cdot \|_1$ denotes the $\ell_1$ norm of a vector and $\odot$ denotes the Hadamard product between vectors. The $j$-th entry of the regularizer vector $\Lambda_i \in \mathbb{R}^d$ is $\lambda_{ij}$. We can write equation (4) in expanded form as:

$$(\forall i \in [g]), \ \hat{w}_i = \arg \min_w w^\intercal \left( \frac{1}{n_i} \sum_{t=1}^{n_i} X_i^t X_i^{t\intercal} \right) w - 2w^\intercal \left( \frac{1}{n_i} \sum_{t=1}^{n_i} y_i^t X_i^t \right) + \sum_{j=1}^d \lambda_{ij} |w_j|, \tag{5}$$

Now, we intentionally replace $\sum_{t=1}^n X_i^t X_i^{t\intercal}$ with $\text{diag}(\sum_{t=1}^n X_i^t X_i^{t\intercal})$. This allows us to write equation (5) as sum of $d$ independent optimization problems:

$$(\forall i \in [g]), \ \hat{w}_i = \arg \min_w \sum_{j=1}^d w_j^2 \left( \frac{1}{n_i} \sum_{t=1}^{n_i} X_{ij}^{t\,2} \right) - 2 \sum_{j=1}^d w_j \left( \frac{1}{n_i} \sum_{t=1}^{n_i} y_i^t X_{ij}^t \right) + \sum_{j=1}^d \lambda_{ij} |w_j|, \tag{6}$$

and subsequently, we get equation (2) as the solution. This is an improper estimator that has the advantage of working well when there are very few samples, i.e., $n_i = \mathcal{O}(1)$ with respect to dimension $d$. It is known that estimating the covariance as needed in the mean squared error requires $n_i$ in $\mathcal{O}(\log d)$ (See Lemma 1 in Ravikumar et al., 2011). Our simple estimator avoids any computation (or estimation) of the covariance matrix which, in any case, would be incorrect if each client has access to only a few samples. Each client $i$ sends the support $S_i$ of $\hat{w}_i$ to the centralized server. Even in the worst-case scenario, each client only sends $d$ bits to the centralized server.

## 4.2 Information aggregation and constructing the final support

We aggregate supports $S_i, \forall i \in [g]$ from all the clients and construct the final support. Before we get to the construction of the final support, we define a random variable $R_{ij}, \forall i \in [g], j \in [d]$ which takes value 1 if $j \in S_i$ and 0 otherwise. Thus, random variable $R_{ij}$ indicates whether entry $j$ is in the support $S_i$ of client $i$. Using the random variables $R_{ij}$, we construct the final support $S$ using majority voting. This is done by computing the median of $R_{ij}$ across $i \in [g]$. If the median is 1 then we conclude that $j$ is in the support otherwise we conclude that $j$ is not in the support. More formally, we define a random variable $R_j \triangleq \frac{1}{g} \sum_{i=1}^g R_{ij}$ and if $R_j \geq \frac{1}{2}$, then we conclude that $j \in S$. Otherwise, if $R_j < \frac{1}{2}$, then we conclude that $j \notin S$. The above procedure can be compactly written as Algorithm 1.

{Part I: Runs in client $i, \forall i \in [g]$}
**Input:** Data samples $(X_i^t, y_i^t), \forall t \in [n_i]$
**Output:** Locally estimated support of shared parameter $w^*$
**for** each $j \in [d]$ **do**
  Compute $\hat{w}_{ij}$ using equation (2) and (3)
  **if** $\hat{w}_{ij} \neq 0$ **then**
    $R_{ij} \leftarrow 1$
  **else**
    $R_{ij} \leftarrow 0$
  **end if**
**end for**
Send $R_{ij}$ to centralized server, $\forall j \in [d]$

{Part II: Runs in centralized server}
**Input:** $R_{ij}, \forall i \in [g], j \in [d]$
**Output:** Globally estimated support $S$ for shared parameter $w^*$
$S \leftarrow \{\}$
**for** each $j \in [d]$ **do**
  Compute $R_j = \frac{1}{g} \sum_{i=1}^{g} R_{ij}$
  **if** $R_j \geqslant \frac{1}{2}$ **then**
    $S \leftarrow S \cup \{j\}$
  **end if**
**end for**

**Algorithm 1:** getExactSupport

## 5 Main Result and Analysis

In this section, we describe and analyze our theoretical results. We present our results in two different settings. In the first setting, we assume that predictor variables are mutually independent. We tackle the more general case of correlated predictors in the second setting. We deal with the special case of correlated Gaussian predictors in Appendix G. Detailed proofs for lemmas and theorems are available in the supplementary material.

### 5.1 Mutually independent predictors

In this setting, predictor variables are mutually independent of each other in all the clients, i.e., $\forall i \in [g]$, $\mathbb{E}(X_{ij}X_{ik}) = 0, \forall j, k \in [d], j \neq k$. In this setting, we state the following result:

**Theorem 1** (Mutually Independent Predictors)**.** *For federated support learning in linear regression, as described in Section 3, with at least $g = \Omega(\log d)$ clients and mutually independent predictor variables if each client has at least $n_i = \Omega(s)$ i.i.d. data samples and the following condition holds:*

$$\max_{i \in [g]} \frac{C}{\sqrt{s}} \left( 8\rho_i^2 \sqrt{\sum_{k \in S^*} w_k^{*2}} + 8|\eta_i\rho_i| \right) < \lambda < \min_{j \in S^*, i \in [g]} |w_j^*(\sigma_{jj}^i)^2| - \frac{C}{\sqrt{s}} \left( 8|w_j^*|\rho_i^2 + 8\rho_i^2 \right.$$
$$\left. \sqrt{\sum_{k \in S^*, k \neq j} w_k^{*2}} + 8|\eta_i\rho_i| \right) \tag{7}$$

*where $C > 0$ is an absolute constant independent of $n_i, s$ and $d$, then Algorithm 1 recovers the exact support of the shared parameter vector $w^*$ with probability at least $1 - \mathcal{O}(\frac{1}{d})$.*

*Proof.* Recall that $R_j = \frac{1}{g} \sum_{i=1}^{g} R_{ij}$ where $R_{ij}$ is defined in Section 4.2. We prove that, with high probability, $R_j \geqslant \frac{1}{2}, \forall j \in S^*$ and $R_j < \frac{1}{2}, \forall j \in S_c^*$. We will provide the proof in two parts. First, we deal with entries $j$ which are in the support of $w^*$, i.e., $j \in S^*$ and then we will deal with $j \in S_c^*$.

**For entries $j$ in support $S^*$.** We begin our proof by first stating the following lemma.

**Lemma 1.** *For $\forall j \in S^*$, let $\mathbb{E}(R_j) > \frac{1}{2}$. With probability at least $1 - 2\exp(-2g(-\frac{1}{2} + \mathbb{E}(R_j))^2 + \log s)$, simultaneously $\forall j \in S^*$, we have $R_j \geqslant \frac{1}{2}$.*

For $g = \Omega(\log d)$, Lemma 1 holds with probability at least $1 - \mathcal{O}(\frac{1}{d})$. Next we show that for any $j \in S^*$, $\mathbb{E}(R_j)$ is indeed greater than $\frac{1}{2}$.

**Lemma 2.** *For $i \in [g]$, $j \in S*$ and some $0 < \delta \leqslant 1$, if predictors are mutually independent of each other and $0 < \lambda_{ij} < |w_j^*(\sigma_{jj}^i)^2| - 8|w_j^*|\rho_i^2\delta - 8\rho_i^2\sqrt{\sum_{k \in S*, k \neq j} w_k^{*2}}\delta - 8|\eta_i\rho_i|\delta$ then we have $\mathbb{E}(R_j) \geqslant 1 - \frac{6}{g}\sum_{i=1}^g \exp(-n_i\delta^2)$.*

In the above lemma, we assume $(\sigma_{jj}^i)^2 > 0$ for all $i \in [g]$ for clarity of exposition. In a more general setting, since $(\sigma_{jj}^i)^2 \geqslant 0$ is the population covariance, it would easy to detect which clients $i$ have $\sigma_{jj}^i = 0$ through the empirical covariance, which would also be zero. Assuming that the proportion of clients for which $\sigma_{jj}^i = 0$ is not too big, one could trivially extend our lemma above by using only the clients $i$ for which $(\sigma_{jj}^i)^2 > 0$.

**For entries $j$ in non-support $S_c^*$.** Similar to the entries in the support, we begin this part by stating the following result for entries in the non-support.

**Lemma 3.** *For $\forall j \in S_c^*$, let $\mathbb{E}(R_j) < \frac{1}{2}$. With probability at least $1 - 2\exp(-2g(\frac{1}{2} - \mathbb{E}(R_j))^2 + \log(d - s))$, simultaneously $\forall j \in S_c^*$, we have $R_j \leqslant \frac{1}{2}$.*

Again, if $g = \Omega(\log d)$, then Lemma 3 holds with probability at least $1 - \mathcal{O}(\frac{1}{d})$. It remains to show that for $\forall j \in S_c^*$, $\mathbb{E}(R_j)$ is smaller than $\frac{1}{2}$. In particular, we use the result from the following lemma.

**Lemma 4.** *For $i \in [g]$, $j \in S_c^*$ and $0 < \delta \leqslant 1$, if predictors are mutually independent of each other and if $\lambda_{ij} > 8\delta\rho_i^2\sqrt{\sum_{k \in S*} w_k^2} + 8|\eta_i\rho_i|\delta$ then we have $\mathbb{E}(R_j) \leqslant \frac{4}{g}\sum_{i=1}^g \exp(-n_i\delta^2)$.*

It is important to note that the statements of Lemma 2 and Lemma 4 are not high probability statements and therefore, a union bound is not required for them. We notice that as long as $\frac{4}{g}\sum_{i=1}^g \exp(-n_i\delta^2) \leqslant \frac{1}{2}$, then $\forall j \in S_c^*, \mathbb{E}(R_j) \leqslant \frac{1}{2}$. Similarly, $\forall j \in S*, \mathbb{E}(R_j) \geqslant \frac{1}{2}$ as long as $\frac{6}{g}\sum_{i=1}^g \exp(-n_i\delta^2) \leqslant \frac{1}{2}$. Results from Lemma 2 and 4 guarantee that the statements of Lemma 1 and 3 hold. Choosing $n_i = \Omega(\frac{1}{\delta^2})$ and $\delta = \frac{C}{\sqrt{s}}, C > 0$, we prove Theorem 1. $\qquad\square$

## 5.2 Correlated predictors

The concentration inequalities used in mutually independent predictors case do not lend themselves directly to the correlated predictors case which makes this analysis more challenging. As described previously, the covariance matrix for $X_i$ is denoted by $\Sigma^i \in \mathbb{R}^{d \times d}$ with diagonal entries $\Sigma_{jj}^i \equiv (\sigma_{jj}^i)^2, \forall j \in [d]$ and non-diagonal entries $\Sigma_{jk}^i \equiv \sigma_{jk}^i, \forall j, k \in [d], j \neq k$. While some of the lemmas from the previous subsection can be reused, we had to come up with some new technical lemmas for this setting. Below, we state the main results for this setting.

**Theorem 2** (Correlated Predictors). *For federated support learning in linear regression, as described in Section 3, with at least $g = \Omega(\log d)$ clients and correlated predictor variables, if each client has $n_i = \Omega(s^2 \log s), s > 1$ i.i.d. data samples and the following condition holds:*

$$\max_{j \in S_c^*, i \in [g]} |\sum_{k \in S*} w_k^*\sigma_{jk}^i| + \frac{C}{s}\left(\sum_{k \in S*} 8\sqrt{2}|w_k^*|(1 + 4\max_j \frac{\rho_i^2}{(\sigma_{jj}^i)^2})\max_j (\sigma_{jj}^i)^2 + 8|\eta_i\rho_i|\right) < \lambda <$$

$$\min_{j \in S*, i \in [g]} |(w_j^*(\sigma_{jj}^i)^2 + \sum_{k \in S*, k \neq j} w_k^*\sigma_{jk}^i)| - \frac{C}{s}\left(8|w_j^*|\rho_i^2 + \sum_{k \in S*, k \neq j} 8\sqrt{2}|w_k^*|(1 + 4\max_j \frac{\rho_i^2}{(\sigma_{jj}^i)^2})\right) \quad (8)$$

$$\max_j (\sigma_{jj}^i)^2 + 8|\eta_i\rho_i|\Bigg)$$

*where $C > 0$ is an absolute constant independent of $n_i, s$ and $d$, then Algorithm 1 recovers the exact support of the shared parameter vector $w^*$ with probability at least $1 - \mathcal{O}(\frac{1}{d})$.*

*Proof.* Recall that $R_j = \frac{1}{g}\sum_{i=1}^g R_{ij}$ where $R_{ij}$ is defined in Section 4.2. We will again prove that, with high probability, $R_j \geqslant \frac{1}{2}, \forall j \in S*$ and $R_j < \frac{1}{2}, \forall j \in S_c^*$. Some of the results from the previous Section 5.1 follow

without any changes. We provide new results for the remaining parts. First, we deal with entries $j$ which are in the support of $w^*$, i.e., $j \in S^*$ and then we will deal with $j \in S_c^*$.

**For entries $j$ in support $S^*$.** We observe that Lemma 1 holds even in this case. Thus, we start our proof by stating the following lemma.

**Lemma 5.** *For $i \in [g]$, $j \in S^*$ and some $0 < \delta \leqslant \frac{1}{\sqrt{2}}$, if $0 < \lambda_{ij} < |(w_j^*(\sigma_{jj}^i)^2 + \sum_{k \in S^*, k \neq j} w_k^* \sigma_{jk}^i)| - 8|w_j^*|\rho_i^2\delta - \sum_{k \in S^*, k \neq j} 8\sqrt{2}|w_k^*|(1 + 4\max_j \frac{\rho_i^2}{(\sigma_{jj}^i)^2})\max_j (\sigma_{jj}^i)^2\delta - 8|\eta_i\rho_i|\delta$ then we have $\mathbb{E}(R_j) \geqslant 1 - \frac{4s}{g}\sum_{i=1}^{g} \exp(-n_i\delta^2)$.*

**For entries $j$ in non-support $S_c^*$.** Again, Lemma 4 follows directly. Thus, we present the following lemma to show that for the entries in the non-support $\mathbb{E}(R_j) < \frac{1}{2}$.

**Lemma 6.** *For $i \in [g]$, $j \in S_c^*$ and some $0 < \delta \leqslant \frac{1}{\sqrt{2}}$, if $\lambda_{ij} > |\sum_{k \in S^*} w_k^* \sigma_{jk}^i| + \sum_{k \in S^*} 8\sqrt{2}|w_k^*|(1 + 4\max_j \frac{\rho_i^2}{(\sigma_{jj}^i)^2})\max_j (\sigma_{jj}^i)^2\delta + 8|\eta_i\rho_i|\delta$ then we have $\mathbb{E}(R_j) \leqslant \frac{4s+2}{g}\sum_{i=1}^{g} \exp(-n_i\delta^2)$.*

The statements of Lemma 5 and Lemma 6 are not high probability statements and therefore, a union bound is not required for them. Note that, as long as we have $(4s+2)\frac{1}{g}\sum_{i=1}^{g} \exp(-n_i\delta^2) < \frac{1}{2}$, we will have $\mathbb{E}(R_j) > \frac{1}{2}, \forall j \in S^*$ and $\mathbb{E}(R_j) < \frac{1}{2}, \forall j \in S_c^*$. Results from Lemmas 5 and 6 ensure that Lemma 1 and 3 hold. Choosing $n_i = \Omega(\frac{1}{\delta^2}\log s)$ and $\delta = \frac{C}{s}, C > 0$, we prove Theorem 2. $\qquad\square$

### 5.3 Time complexity

Each client does $\mathcal{O}(dn_i)$ basic calculations. Thus, the time complexity at each client is $\mathcal{O}(sd)$ for mutually independent predictors and $\mathcal{O}(s^2 d \log s)$ for correlated predictors. The centralized server gathers $d$-bits of information from $g$ clients in $\mathcal{O}(dg) = \mathcal{O}(d \log d)$ time.

## 6 Choice of Regularizer

The conditions mentioned in Equations (7) and (8) provide sufficient theoretical conditions on the regularizers $\lambda_{ij} = \lambda$ for exact support recovery. It remains to be shown whether such a choice of $\lambda$ is feasible. If we let $\min_{j \in S^*} \sigma_{jj}^i = \sigma_i^l$, then it can be shown that equation (7) has a feasible solution as long as we choose a $C$ such that $\frac{C}{\sqrt{s}} < \frac{w^l \sigma_i^{l^2}}{16\rho_i^2 w^h \sqrt{s} + 8w^h \rho_i^2 + 16\eta_i\rho_i}$. This setting of $\lambda$ in equation (7) can be contrasted with the setting of the regularizer $\lambda_{\text{lasso}}$ of a centralized-server lasso problem in Theorem 1 of Wainwright (2009b). In particular, the lower bound on $\lambda$, $\mathcal{O}(1) + \mathcal{O}(\frac{1}{\sqrt{n_i}})$, is analogous to the lower bound on the centralized-server lasso regularizer $\lambda_{\text{lasso}}$, i.e., $\lambda_{\text{lasso}} > \mathcal{O}(\sqrt{\frac{\log d}{\sum_{i=1}^{g} n_i}})$. However, our choice of $\lambda$ is independent of $d$. In Section 8, we empirically validate the existence of a feasible range for $\lambda$. A similar analysis can be carried out for the feasibility of equation (8). Let $M_{AB} \in \mathbb{R}^{|A| \times |B|}$ be a matrix constructed by restricting rows of $M$ to entries in $A$ and columns of $M$ to entries in $B$ for two sets $A, B \subseteq [d]$. Furthermore, for a matrix $M \in \mathbb{R}^{p \times q}$, let $\|M\|_\infty \triangleq \max_{i \in [p]} \sum_{j \in [q]} |M_{ij}|$. We assume that $\|\Sigma_{S_c^* S^*}^i\|_\infty + \|\Sigma_{S^* S^*}^i\|_\infty < \gamma$ for some constant $\gamma > 0$. We further assume that $\gamma$ is so small such that $w^l \sigma_i^{l^2} - w^h\gamma > 0$. This assumption is similar to the *incoherence* assumption in Wainwright (2009b) and it ensures that predictors within the support are not highly correlated, as well as predictors outside the support do not exert a high influence on the predictors within the support. For ease of notation, we will denote the term $(1 + 4\max_r \frac{\rho_i^2}{(\sigma_{rr}^i)^2})\max_r (\sigma_{rr}^i)^2$ by a positive constant $k_5$. We observe that if $\frac{C}{s} < \frac{w^l \sigma_i^{l^2} - w^h\gamma}{16\sqrt{2}k_5 s + 8w^h \rho_i^2 + 16\eta_i\rho_i}$, then the choice of $\lambda$ in equation (8) is feasible. Notice how the upper bound $\lambda < w^l \sigma_i^{l^2} - w^h\gamma > 0$ is similar to a combination of the minimum weight condition and the incoherence condition from Wainwright (2009b). Similarly, the lower bound $\lambda > \mathcal{O}(1) + \mathcal{O}(\sqrt{\frac{\log s}{n_i}})$ is analogous to the setting of the centralized-server lasso regularizer in Wainwright (2009b). We provide the

following illustrative example to explain the similarity between technical conditions required for the feasibility of equation (8) and the standard mutual incoherence condition.

## 6.1 An illustrative example

Consider the data generation process given in equation (1), i.e., $y = X^T w^* + e$. We have dropped the subscript $i$ for brevity as the following argument holds for any client $i \in [g]$. We take $d = 3$, the parameter vector $w^* \in \mathbb{R}^3$ looks like $w^* = \begin{bmatrix} a \\ a \\ 0 \end{bmatrix}$ where $a > 0$, i.e., $S^* = \{1, 2\}$ and $S_c^* = \{3\}$. We assume that entries of the design matrix $X_k$ are in $\{-1, 1\} \forall k = 1, 2, 3$. Furthermore, let $\mathbb{E}(X_1 X_2) = q$, $\mathbb{E}(X_2 X_3) = p$, $\mathbb{E}(X_1 X_3) = 0$, $\mathbb{E}(X_1) = 0$ and $\mathbb{E}(X_3) = 0$, where $\mathbb{E}(.)$ denotes the expected value.

Let $\Sigma$ be $\mathbb{E}(X^T X)$ where $X = \begin{bmatrix} X_1 \\ X_2 \\ X_3 \end{bmatrix} \in \{-1, 1\}^3$. Then,

$$\Sigma = \begin{bmatrix} 1 & q & 0 \\ q & 1 & p \\ 0 & p & 1 \end{bmatrix}$$

Thus, the feasibility criteria for $\lambda$ in equation (8) becomes: $|w_1^* \sigma_{31} + w_2^* \sigma_{32}| + A(w*, \Sigma, \rho, \eta)\frac{C}{s} < \frac{C}{s} \min(|w_1^* \sigma_{11}^2 + w_2^* \sigma_{12}| + B(w*, \Sigma, \rho, \eta), |w_2^* \sigma_{22}^2 + w_1^* \sigma_{21}| + D(w*, \Sigma, \rho, \eta))$,

where $A(w^*, \Sigma, \rho, \eta), B(w^*, \Sigma, \rho, \eta)$ and $D(w^*, \Sigma, \rho, \eta)$ are upper bounded by $O(s)$ and thus these products can be made arbitrarily small by choosing an appropriate $C$. In the worst-case scenario, the above equation becomes (by replacing $w^*$ and $\Sigma$):

$$a|p| < a - a|q| + E(w^*, \Sigma, \rho, \eta)\frac{C}{s}$$

Again, $E(w^*, \Sigma, \rho, \eta)\frac{C}{s}$ can be made arbitrarily small by choosing an appropriate $C$. The above feasibility criteria simplifies to :

$$|p| + |q| < 1 + \epsilon, \tag{9}$$

where $\epsilon$ is an arbitrarily small quantity. We compare this with the mutual incoherence condition from Wainwright (2009b) which requires $\|\Sigma_{S_c^* S^*} \Sigma_{S^* S^*}^{-1}\|_\infty \leqslant 1 - \alpha$, for some $\alpha \in (0, 1]$. This simplifies to

$$\left\| \begin{bmatrix} 0 & p \end{bmatrix} \begin{bmatrix} 1 & q \\ q & 1 \end{bmatrix}^{-1} \right\|_\infty < 1$$

which is equivalent to

$$|p| + |q| < 1 . \tag{10}$$

Notice how for small values of $p$ and $q$ (say $p = 0.2$, $q = 0.1$) both equation (9) and (10) are easy to satisfy while for the high value of $p$ and $q$ (say $p = 0.7$, $q = 0.6$) both conditions fail to hold.

## 7 Discussion on Robustness

Since our method only relies on the correct calculation of the median, it is naturally robust to the failure of some clients. To simulate the effect of model poisoning (Bhagoji et al., 2019) and stragglers, we consider that

a proportion $0 < \beta < \frac{1}{2}$ of clients have gone rogue (are straggling) and transmitting the wrong information to the centralized server. For the worst-case scenario, we assume that they report the complement of the support, i.e., they always send a bit "1" for entries in the non-support and a bit "0" for entries in the support. To accommodate this change in the case of correlated predictors, we slightly change statements of Lemmas 5 and 6. Now we have, $(\forall j \in S^*), \quad \mathbb{E}(R_j) \geqslant (1 - \beta) - \frac{4s}{g} \sum_{i=1}^{(1-\beta)g} \exp(-n_i \delta^2)$ and $(\forall j \in S_c^*), \quad \mathbb{E}(R_j) \leqslant \frac{4s+2}{g} \sum_{i=1}^{(1-\beta)g} \exp(-n_i \delta^2) + \beta$. It is easy to see that, as long as, we have $n_i > \frac{1}{\delta^2} \log(\frac{(8s+4)(1-\beta)}{1-2\beta})$ data samples per client, then we still have $\mathbb{E}(R_j) > \frac{1}{2}, \forall j \in S^*$ and $\mathbb{E}(R_j) < \frac{1}{2}, \forall j \in S_c^*$ and all our results still hold. Similarly, for mutually independent predictors, our results hold as long as we have $n_i > \frac{1}{\delta^2} \log(\frac{12(1-\beta)}{1-2\beta})$.

## 8 Validating Theory with Synthetic Experiments

In this section, we validate our theoretical results by conducting computational experiments.

We provide the results for the experiments when predictors are correlated. Data in each client is generated by following the generative process described in equation 1. Note that predictors and error term in different clients follow different sub-Gaussian distributions. To make it more general, we keep the correlation between entries in the support different than the correlation between one entry in the support and the other entry in the non-support and these further vary across clients. The regularization parameter $\lambda_i$ for each client is chosen such that the condition in Theorem 2 is satisfied for every client. All the results reported here are averaged over 30 independent runs. We conduct two separate experiments to verify that $n_i = \Omega(s^2 \log s)$ independent samples per client and a total of $g = \Omega(\log d)$ clients are sufficient to recover the true support.

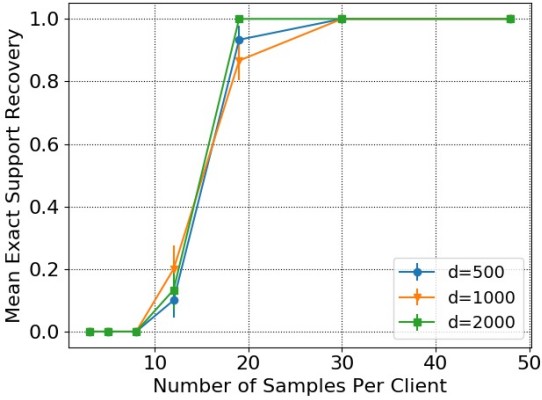
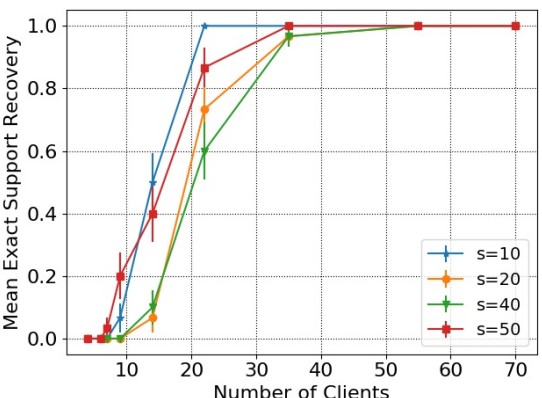

(a) Exact support recovery against numbers of samples per client

(b) Exact support recovery against numbers of clients

Figure 2: Phase transition curves. Left: Exact support recovery averaged across 30 runs against varying number of samples per client for dimension $d = 500, 1000$, and 2000, sparsity $s = 3$, $g = \Omega(\log d)$ clients. Right: Exact support recovery averaged across 30 runs against varying number of clients for sparsity $s = 10, 20, 40$, and 50, dimension $d = 1000$, $n = \max(30, \Omega(s^2 \log s))$ samples per server.

### 8.1 Exact support recovery against number of samples per client

This experiment was conducted for a varying number of predictors ($d = 500, 1000$, and 2000). For each of them, we fixed the number of clients to be $g = 2 \log d$. The sparsity $s$ is kept fixed at 3. The number of samples per client $n_i$ is varied with control parameter $C$ as $10^C s^2 \log s$. The performance of our method is measured by assigning a value 1 for exact recovery and 0 otherwise. We can see in Figure 2a, that initially, recovery remains at 0 and then there is a sharp jump after which recovery becomes 1. Notice how all three curves align perfectly. This validates the result of our theorem and shows that given $g = \Omega(\log d)$ clients, $n_i = \Omega(s^2 \log s)$ samples per client are sufficient to recover the true support.

### 8.2 Exact support recovery against number of clients

The second experiment was conducted for a varying number of non-zero entries ($s = 10, 20, 40,$ and $50$) in the support of $w^*$. The experiments were run for a setup with $d = 1000$ predictors. We fixed the number of samples per client ($n_i$) to be $\max(30, \Omega(s^2 \log s))$. This ensures that a minimum of 30 samples are available to each client. This is inline with our previous experiment where exact recovery is achieved around 30 samples per client. The number of clients $g$ is varied with control parameter $C$ as $10^C \log d$. Like the previous experiment, performance is measured by assigning a value 1 for exact recovery and 0 otherwise. We can again see in Figure 2b, that initially, recovery remains at 0 and then it goes to 1 as we increase the number of clients. We also notice that all four curves align nicely. This validates that given $n_i = \Omega(s^2 \log s)$ independent samples per server, $g = \Omega(\log d)$ clients are sufficient to recover the true support.

### 8.3 Robustness to straggling clients

Recall that since our method only relies on the correct calculation of the median, it is naturally robust to the failure of some clients. Our next experiment simulates the effect of having a proportion $0 < \beta < \frac{1}{2}$ of straggling clients which transmit the wrong information to the centralized server. For the worst-case scenario, we assume that they report the complement of the support, i.e., they always send a bit "1" for entries in the non-support and a bit "0" for entries in the support. Table 2 shows that our method is robust for a proportion $0 < \beta < 0.3$ of straggling clients.

Table 2: Exact support recovery averaged across 30 runs for different proportions $\beta$ of straggling clients for dimension $d = 1000$, sparsity $s = 3$, $g = \Omega(\log d)$ clients, $n_i = \Omega(s^2 \log s)$ samples per client.

| $\beta$ | 0.10 | 0.20 | 0.30 | 0.35 | 0.40 |
|---|---|---|---|---|---|
| Mean exact support recovery | 100% | 100% | 100% | 80% | 33% |

### 8.4 Comparison with centralized lasso

We compared our method to the centralized-server lasso Wainwright (2009b), which has access to all the data, unlike our method. Table 3 shows that both our method and centralized-server lasso are successful in recovering the true support, but our method requires less computation.

Table 3: Exact support recovery and runtime averaged across 30 runs for dimension $d = 500, 1000,$ and $2000$, sparsity $s = 3$, $g = \Omega(\log d)$ clients, $n_i = \Omega(s^2 \log s)$ samples per client. For easy comparison, runtime was normalized with respect to our method for $d = 500$.

| $d$ | Mean exact support recovery | | Mean runtime | |
|---|---|---|---|---|
| | Our method | Centralized lasso | Our method | Centralized lasso |
| 500 | 100% | 100% | 1 | 4.1 |
| 1000 | 100% | 100% | 2.4 | 11.6 |
| 2000 | 100% | 100% | 4.6 | 26.7 |

## 9 Real World Experiment

In this section, we demonstrate the effectiveness of our method to determine the support of a sparse linear regression setup in a real world data set. We used the BlogFeedback data set (Buza, K., 2014) from `https://archive.ics.uci.edu/ml/datasets/BlogFeedback`. This data set contains features extracted from blog posts and the task is to predict how many comments the post will receive using these features. We divided data into training and test data by choosing 80% of all samples to be training data at random. The details about the data set are as follows:

- Number of training samples: 41917

- Number of test samples: 10480

- Number of features (after removing all zeros columns): 276

## 9.1 Comparing recovered support with centralized lasso

Since the true support of the parameter vector is unknown for real-world data, we constructed a "centralized" support for comparison by running lasso on the complete training data set. This simulates a centralized server with access to all the data. The "centralized" support contains 42 elements. We want to compare the support recovered from our method, called "federated" support, with this 'centralized' support.

## 9.2 Performance measures

We defined the following performance measure for comparison:

$$\text{Jaccard Index} = \frac{\text{Number of common elements in the "federated" support and in the "centralized" support}}{\text{Number of elements in the union of "centralized" support and "federated" support}} \quad (11)$$

## 9.3 Case 1

For the first experiment, we divided the dataset randomly into 419 clients with each client containing 100 samples (except the last one which contains more to account for imbalance). This is a highly distributed setting where each client has access to a small number of samples. We conducted our experiment using $\lambda = 0.08$. Our method recovered support with 48 elements, with a Jaccard Index of 0.76.

To compare the generalization on the test data set, we computed parameter vectors $w_{\text{fed}}$ and $w_{\text{cen}}$ by running simple linear regression on training samples restricted to "federated" support and "centralized" support respectively. After that, Mean Square Error (MSE) is computed on the test samples using the recovered $w_{\text{fed}}$ and $w_{\text{cen}}$. We observed that MSE for our method, 0.71545, is slightly better than MSE for "centralized" support, 0.71615.

## 9.4 Case 2

For the second experiment, we divided the dataset randomly into 41 clients with each client containing 1000 samples (except the last one which contains more to account for imbalance). This is a setting where we have only a few clients. We again conducted our experiment using $\lambda = 0.08$. Our method recovered support with 49 elements, with a Jaccard Index of 0.8.

Again, we followed the same procedure as Case 1 to compute MSE in test samples. We observed that MSE for our method, 0.70525, is slightly better than MSE for "centralized" support, 0.70548.

We see that in both cases, our method recovered a similar set of support as the support recovered by lasso running in a centralized server. Our method also generalizes to the test dataset in a similar manner, even performing better in terms of MSE.

Table 4: Jaccard index for different regularizer values $\lambda$.

| $\lambda$ | 0.04 | 0.06 | 0.08 | 0.10 | 0.12 | 0.14 | 0.16 | 0.18 | 0.20 |
|---|---|---|---|---|---|---|---|---|---|
| Case 1 | 0.46 | 0.83 | 0.76 | 0.73 | 0.76 | 0.77 | 0.63 | 0.53 | 0.42 |
| Case 2 | 0.56 | 0.71 | 0.80 | 0.82 | 0.75 | 0.78 | 0.83 | 0.74 | 0.76 |

### 9.5 Robustness to the choice of regularizer

To evaluate the robustness of our method to the choice of regularizer $\lambda$, we tested a range of regularizer values in the real-world data. Table 4 shows that our method is relatively robust in a wide range of regularizer values (i.e., $0.06 \leqslant \lambda \leqslant 0.14$).

## 10 Concluding Remark

In this paper, we propose a simple and easy-to-implement method for learning the exact support of parameter vector of linear regression problem in a federated learning setup. We provide theoretical guarantees for the correctness of our method. We also show that our method runs in polynomial sample and time complexity. Furthermore, our method is averse to client failures, model poisoning, and straggling clients.

As a future direction, it would be interesting to apply our ideas to other estimation problems involving sparse regression such as non-parametric regression (Ravikumar et al., 2007), learning probabilistic graphical models (Ravikumar et al., 2010) and diffusion networks (Daneshmand et al., 2014). These problems are often handled in the centralized setting but it would be interesting to tackle them in the federated setting without compromising on the performance.

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
