# OpenReview forum: "Recovering Exact Support in Federated lasso without Optimization"
_TMLR — Accepted by TMLR_

### Review · Reviewer_sv9G · 2023-10-09

**Summary Of Contributions:**

The paper considers the problem of learning the support of a distribution over a distributed set of agents. The novelty of this paper resides in the analysis of the convergence results and complexities provided. The paper showcases the method in a real-world dataset.

**Audience:**

Yes

**Broader Impact Concerns:**

None.

**Claims And Evidence:**

Yes

**Requested Changes:**

# Minor Comments

**Lambda -** It seems to me that  Lambda depends on quantities that cannot be evaluated in practice. As it is common in ML, the value of Lambda can be obtained empirically, but how sensitive the algorithm is to the correct choice of Lambda is unclear to me. The experiments section indicates that Lambda=0.08 was used, but nothing is said of why this value has been chosen or how it has been selected. I would suggest the authors consider providing more information regarding the choice of lambda in practice.

**Modifications**

“It is obvious that for” → “For “

**Strengths And Weaknesses:**

# Strengths

**S1** - The paper presents a simple and clear algorithm to obtain the exact support for the federated Lazzo problem that requires little computation. The advantage of this method relies on its simplicity and its theoretical guarantees of convergence.

**S2** - The paper presents a numerical experiment with real-world data that validates the theoretical claims put forward by the authors.

S3 - The authors ‘one-shot communication’ framework makes the method widely applicable in practice. Compounded on the very low computations required, this method seems very well suited for a wide range of IOT applications.

# Weaknesses

**W1** - The paper seems to cover a very limited scenario where the data is generated according to a linear model. Even though these results are important to understanding federated learning under Lasso regularization, the results provided in this work have very limited applicability in practice.

**W2** - The paper’s structure makes it difficult to understand. Even though Lasso is a widely known problem, the exact problem formulation is never explicitly written. This makes the first read of the paper rough and unclear. I would suggest explicitly writing the problem formulation.

**W3** - The choice of a single Lambda for all agents seems like an arbitrary solution that is key in the method but that is not explained. The authors say:

*“we will present a more practical choice of a single lambda across all clients and entries”*

What are the implications of this decision and what is the intuition for it?

---

> ### Author Response · Authors · 2023-11-27
>
> > S1. advantage... simplicity and theoretical guarantees
> >
> > S2. experiment with real-world data... validates theoretical claims
> >
> > S3. 'one-shot communication' framework... widely applicable... low computation... well suited for IOT applications
>
> We thank the reviewer for the appreciation of our work.
>
> > W1. limited scenario... linear model. Even though these results are important... limited applicability
>
> We illustrate our novel idea (one-shot communication framework) with a very relevant problem: sparse linear regression. We hope this will motivate work on "other estimation problems involving sparse regression such as non-parametric regression [Ravikumar'07], learning probabilistic graphical models [Ravikumar'10] and diffusion networks [Daneshmand'14]. These problems are often handled in the centralized setting but it would be interesting to tackle them in the federated setting without compromising on the performance." (See Section 10.)
>
> > W2. structure makes it difficult to understand... problem formulation is never explicitly written
>
> Our method uses a closed-form solution of an "improper" lasso. This is the strength of our method, as we explain next.
>
> As mentioned in Section 1: "the number of clients could be quite large, each client is typically a simple device that has access to a very small number of samples and can only conduct very basic computations due to limitations on its processing and power capabilities. Furthermore, since battery power is at a premium, the communication between the client and the centralized server acts as a major bottleneck."
>
> Consider for instance each client having only 2 samples. In this case, solving the original lasso optimization problem on each client is not only power-consuming but also useless, since solving the original lasso from only 2 samples will almost surely lead to a wrong estimated parameter vector, and thus, failure in support recovery.
>
> Please see eq.(4) containing the original lasso problem which is equivalent to eq.(5) containing the covariance matrix. In order to derive eq.(6), we intentionally replace the covariance matrix with the diagonal of the covariance matrix. As mentioned in Section 4.1: "This is an improper estimator that has the advantage of working well when there are very few samples, i.e., $n_i$ is $O(1)$ with respect to dimension $d$. It is known that estimating the covariance as needed in the mean squared error requires $n_i \in O(\log d)$ (See Lemma 1 in Ravikumar et al., 2011). Our simple estimator avoids any computation (or estimation) of the covariance matrix which, in any case, would be incorrect if each client has access to only a few samples."
>
> Due to the replacement of the covariance by its diagonal matrix in eq.(6), we arrive to a closed-form solution in eq.(2) and eq.(3). Thus, each client computes its parameter vector very easily, and submits its "support guess" (non-zero entries of the parameter vector) to the central server (See Algorithm 1, left). Then, the central server collects all the "support guesses" and uses the median to compute the final support. (See Algorithm 1, right).
>
> > W3. "single lambda across all clients and entries" implications?... intuition?
>
> Please see Section 6 (Choice of Regularizer) for a discussion on this topic. In addition, the work on centralized-server Lasso (See e.g., [Wainwright, 2009a] and follow ups) uses a single lambda across all entries.
>
> > Minor: Lambda depends on quantities that cannot be evaluated in practice... In ML... Lambda can be obtained empirically... experiments: Lambda=0.08... why
>
> $w^*,\sigma,\rho$ are important parameters (IPs) that somehow define the problem complexity.
>
> Setting the regularization parameter $\lambda$ from IPs was done in several other works, which clearly did not prevent them from being published. To mention a few:
> - linear regression [Wainwright'09, Sharp Thresholds for High-Dimensional and Noisy Sparsity Recovery Using L1-Constrained QP]
> - learning Ising models [Ravikumar'10, High-Dimensional Ising Model Selection using L1-Regularized Logistic Regression]
> - learning Gaussian graphical models [Ravikumar'11, High-dimensional Covariance Estimation by Minimizing L1-penalized Log-determinant Divergence]
> - nonparametric regression [Ravikumar'07, Sparse Additive Models]
> - diffusion networks [Daneshmand'14, Estimating Diffusion Network Structures]
>
> As in the above papers, IPs' exact values are not needed, and only reasonable lower and upper bounds are enough to set $\lambda$. (See Section 6.)
>
> The effect of varying $\lambda$ in the real-world experiment:
> | $\lambda$ | Jaccard index Case 1 | Case 2 |
> | --- | --- | --- |
> | .04 | .46 | .56 |
> | .06 | .83 | .71 |
> | .08 | .76 | .8 |
> | .10 | .73 | .82 |
> | .12 | .76 | .75 |
> | .14 | .77 | .78 |
> | .16 | .63 | .83 |
> | .18 | .53 | .74 |
> | .2 | .42 | .76 |
>
> We will make the other proposed minor corrections.

---

### Review · Reviewer_y1zc · 2023-10-13

**Summary Of Contributions:**

This paper presents a new distributed algorithm for support recovery for LASSO in the federated setting where individual users have different data distributions. The algorithm is shown to have low computational cost for each of the users and requires a simple majority vote at the central server, near-optimal statistical complexity (however, requiring that every user has $\widetilde{\Omega} (s)$ samples in the uncorrelated case, and $\widetilde{\Omega} (s^2)$ samples in the correlated case), and is also robust. The algorithm is theoretically shown to work when the individual user covariances are diagonal and non-diagonal. The theoretical results are rounded out with an experimental analysis of the algorithm.

**Audience:**

Yes

**Broader Impact Concerns:**

None.

**Claims And Evidence:**

Yes

**Requested Changes:**

See strengths/weaknesses.

**Strengths And Weaknesses:**

The paper presents a new distributed algorithm for LASSO in the federated setting. The main point I take away from this paper is that the algorithm is computationally very efficient and has a very small communication footprint (only requiring a one-shot transmission of a support). I think the results in the paper are interesting, both to the federated learning community, as well as for designing better algorithms for LASSO, as the algorithm can be run as a parallel algorithm on $\log(d)$ threads on a single server to solve the LASSO problem. It would be very interesting to add in a section in the paper comparing the computational cost of this algorithm with other state-of-the-art fast algorithms for LASSO in the centralized setting. The communication footprint of the algorithm is very small, and I believe the analysis can be improved since each user returns a support size which is also $\widetilde{O} (s)$ with high probability (requires a proof), in which case, the support of each user can be communicated in $\widetilde{O} (s \log(d))$ bits by transmitting their locations. I believe this should be asymptotically optimal. Since the algorithm aggregates the individual supports by a majority rule, existing approaches can be used to show that the algorithm is robust to stragglers, as the authors show. This is nice, as in the federated setting, individual devices may drop in and out of the network, and may be subject to differing latencies (stragglers), or be malicious.

In general, looking at the analysis in the paper for the uncorrelated case, it appears that as long as $\sigma_{jj}^i$ is not too small for any $(i,j)$, and $w_i$ is not too small, by choosing a sufficiently small value of $\delta$, the algorithm's correctness can be established.
However, in this case, it is confusing to me why the algorithm fails when $\sigma_{jj}^i = 0$ for some $i$. In this case, even though the support is not-identifiable from the data present with user $i$, since this user's data cannot be used to tell whether $w_j$ is or isn't $0$, the presence of other users with $\sigma_{jj}^i \ne 0$ should allow the support to be identifiable. I believe that the current algorithm should still succeed in this case, and this follows from the same reasons that the algorithm is robust. This allows the minimum over all $i \in [g]$ to be changed into the $(1-\beta)^{th}$ percentile smallest minimum in the upper bound on $\lambda$. In the limiting case where $\sigma_{jj}^i = 0$ for all $i$ and some $j$, the current analysis seems to be tight.

The picture seems more confusing in the correlated case. I fail to find a non-trivial example of parameters where eq. (8) holds. Notice that the lower bound on $\lambda$ is of the form $\max_{j \in S_c^\star, I \in [g]} | \sum_{k \in S^\star} w_k^\star \sigma_{jk}^i| + \cdots$, while the upper bound is of the form $\min_{j \in S^\star, i \in [g]} |w_j^\star (\sigma_{jj}^i)^2 + \sigma_{k \in S^\star, k\ne j} w_k^\star \sigma_{jk}^i| - \cdots$, where the $\cdots$ terms contain all the dependency on $\delta$, and by extension, $n_i$. The $\min$ in the upper bound can be relaxed to $\min_{j \in S^\star_c}$. However, even then, if $\sigma_{jj}^i < 1$ for any $i$, the upper and lower bounds on $\lambda$ contradict each other and correctness can no longer be established for the algorithm, let alone finding a suitable value of $\lambda$. This indicates to me that the analysis in the correlated case might be loose. I do not have an intuition about whether the algorithm might fail in the case (although the experiments seem to not suggest this possibility).

Overall, I think the contribution in the paper is nice, however, not the tightest possible. I would have appreciated it if the authors spent some more effort in making the analysis in the paper tighter, and providing some more insights about when the bounds work, and when they fail. It's nice to see an experimental section and a study into the practical performance of the algorithm. The writing of the paper is ok, but the appendix can be improved quite significantly. It would help to include a little bit of structure as to what lemmas prove what results, and reduce the number of lemmas (since many of them seem to repeat). I would appreciate if the authors put some effort into this, as the current version of the appendix is not up to publication standard in my opinion.

Minor:
1. $n_i = o(s \log(d))$ is not the same as $n_i \le k s \log(d)$ for some constant $k > 0$.
2. In the paragraph right before 4.2 there is a typo: $n_i$ is independent of $O(1)$ with respect of dimension $d$.
3. Use a different notation than $e_i$ for noise (eg. Z_i). It's easy for a reader to confuse this with the standard basis vectors.
4. How are Lemmas 10 and 11 different from Lemmas 8 and 9? The statements are identical? (same for Lemmas 12 and 13). It would be nice to streamline the appendix and remove bloat in the form of repeating lemmas.

---

> ### Author Response · Authors · 2023-11-27
>
> > results... are interesting, both to the federated learning community, as well as for designing better algorithms for LASSO, as the algorithm can be run as a parallel algorithm on $\log(d)$ threads on a single server to solve the LASSO problem
>
> We thank the reviewer for the appreciation of our work.
>
> > It would be very interesting to add in a section in the paper comparing the computational cost of this algorithm with other state-of-the-art fast algorithms for LASSO in the centralized setting
>
> We compared to the centralized-server Lasso (Wainwright 2009b). Sparsity $s=3$, $g=\Omega(\log d)$ clients, $n_i=\Omega(s^2\log s)$ samples per client.
> | $d$ | our mean time (30 runs) | lasso |
> | --- |--- | --- |
> | 500 | 1 | 4.1 |
> | 1000 | 2.4 | 11.6 |
> | 2000 | 4.6 | 26.7 |
>
> We kindly remind that TMLR guidelines for reviewers specifically state "it should not be used as a reason to reject work ... because it isn't achieving a new state-of-the-art on some benchmark." We kindly refer to https://jmlr.csail.mit.edu/tmlr/reviewer-guide.html
>
> > communication footprint... is very small... analysis can be improved since each user returns a support size which is also $\tilde{O}(s)$ with high probability (requires a proof), in which case... can be communicated in $\tilde{O}(s \log(d))$ bits by transmitting their locations.
>
> Proving that the support of every client has size $\tilde{O}(s)$ could not be possible. Unfortunately, we work on a setting where the number of samples for each client $i$ is very small, i.e., $n_i$ grows strictly slower than $s\log d$. This prevents any claim of support correctness on each client, or even any claim of support size on each client. Using the median is what allows any sort of correctness and thus, support size. (Recall that the central server collects all the "support guesses" and uses the median to compute the final support.)
>
> > for the uncorrelated case... why the algorithm fails when
> $\sigma^i_{jj}=0$ for some $i$... even though the support is not-identifiable from the data present with user $i$... the presence of other users with $\sigma^i_{jj}\neq 0$ should allow the support to be identifiable... the current algorithm should still succeed... this follows from the same reasons that the algorithm is robust. This allows the minimum over all $i \in [g]$ to be changed into the $(1-\beta)^{th}$ percentile smallest minimum in the upper bound on $\lambda$...
>
> ${\sigma^i_{jj}}^2$ is the population variance for client $i$, feature $j$.
> We agree with the reviewer, we could add $\sigma^i_{jj}\neq 0$ to the condition $i\in [g]$, and assume that the proportion for which $\sigma^i_{jj}=0$ is not too big.
>
> > in the correlated case. I fail to find a non-trivial example of parameters where eq.(8) holds... The $\min$ in the upper bound can be relaxed to $\min_{j\in S^*_c}$. However... the upper and lower bounds on $\lambda$ contradict each other...
>
> Please see Section 6 (Choice of Regularizer) for a discussion on this topic. The $\min$ in the upper bound cannot be changed from $\min_{j\in S^*}$ to $\min_{j\in S^*_c}$. The minimums are over two different sets (a set and its complement).
>
> > however, requiring that every user has $\tilde{\Omega}(s)$ samples in the uncorrelated case, and $\tilde{\Omega}(s^2)$ samples in the correlated case
> >
> > Overall, I think the contribution in the paper is nice, however, not the tightest possible. I would have appreciated it if the authors spent some more effort in making the analysis in the paper tighter, and providing some more insights about when the bounds work, and when they fail.
> >
> > I would appreciate if the authors put some effort into this, as the current version of the appendix is not up to publication standard in my opinion.
>
> We hope our answers above might change the reviewer's perception. In addition, please see Section 6 (Choice of Regularizer) and 6.1 (An Illustrative Example) which includes, among other things, a discussion of the standard mutual incoherence condition in the centralized-server Lasso (Wainwright 2009b).
>
> > The writing of the paper is ok, but the appendix can be improved... what lemmas prove what results, and reduce the number of lemmas
> >
> > Minor: Lemmas 10 and 11 different from Lemmas 8 and 9?... same for Lemmas 12 and 13
>
> Lemmas 8, 9, 12 are for $j\in S^*$, Lemmas 10, 11, 13 are for $j\in S^*_c$.
>
> > Minor: $n_i=o(s\log d)$ not the same as $n_i<ks\log d$ for some constant $k>0$
>
> We will rephrase with $n_i=o(s\log d)$. Our intention was to say that the number of samples $n_i$ grows strictly slower than $s\log d$.
>
> We will make the other proposed minor corrections.

---

### Review · Reviewer_w6rm · 2023-11-06

**Summary Of Contributions:**

This paper enables federated learning for the LASSO problem. In particular, it shows how to optimize this model and provides theoretical analysis for the convergence rates and statistical bounds.

**Audience:**

No

**Claims And Evidence:**

Yes

**Requested Changes:**

1. This paper claims the proposed algorithm has no optimization for finding the optimal solution. This is not true. In fact, it uses the soft-thresholding approach to solve the lasso problem. Thus, it overclaims the contribution.

2. The proposed algorithm is weird. It seems there is only one iteration. However, the standard soft-thresholding approach is an iterative approach, which requires many iterations. Why does the proposed algorithm just need one iteration?

3. What unique challenges are there in the theoretical analysis compared with standard lasso problems?

**Strengths And Weaknesses:**

1. The writing is good. It is easy to follow.

2. This paper provides a theoretical analysis for the federated lasso problem.

---

> ### Author Response · Authors · 2023-11-17
>
> > This paper... provides theoretical analysis for the convergence rates...
> >
> > Change 1. This paper claims the proposed algorithm has no optimization... This is not true... it uses the soft-thresholding approach to solve the lasso problem.
> >
> > Change 2. The proposed algorithm is weird. It seems there is only one iteration.
>
> Thanks for the opportunity to clarify our main motivation. We do not provide convergence rates in the optimization sense. Our method uses a closed-form solution of an "improper" lasso, which makes it look like as a "single iteration". This is the strength of our method, as we explain next.
>
> As mentioned in Section 1: "the number of clients could be quite large, each client is typically a simple device that has access to a very small number of samples and can only conduct very basic computations due to limitations on its processing and power capabilities. Furthermore, since battery power is at a premium, the communication between the client and the centralized server acts as a major bottleneck."
>
> Consider for instance each client having only 2 samples. In this case, solving the original lasso optimization problem on each client is not only power-consuming but also useless, since solving the original lasso from only 2 samples will almost surely lead to a wrong estimated parameter vector, and thus, failure in support recovery.
>
> Please see eq.(4) containing the original lasso problem which is equivalent to eq.(5) containing the covariance matrix. In order to derive eq.(6), we intentionally replace the covariance matrix with the diagonal of the covariance matrix. As mentioned in Section 4.1: "This is an improper estimator that has the advantage of working well when there are very few samples, i.e., $n_i$ is $O(1)$ with respect to dimension $d$. It is known that estimating the covariance as needed in the mean squared error requires $n_i \in O(\log d)$ (See Lemma 1 in Ravikumar et al., 2011). Our simple estimator avoids any computation (or estimation) of the covariance matrix which, in any case, would be incorrect if each client has access to only a few samples."
>
> Due to the replacement of the covariance by its diagonal matrix in eq.(6), we arrive to a closed-form solution in eq.(2) and eq.(3). Thus, each client computes its parameter vector very easily, and submits its "support guess" (non-zero entries of the parameter vector) to the central server (See Algorithm 1, left). Then, the central server collects all the "support guesses" and uses the median to compute the final support. (See Algorithm 1, right).
>
> > Change 3. What unique challenges are there in the theoretical analysis compared with standard lasso problems?
>
> The whole theoretical analysis is entirely novel. We are not aware of any other paper that has tried our approach of avoiding using an optimization algorithm. Thus, our paper required a novel way to perform the technical analysis.
>
> Reviewer y1zc states "... the results in the paper are interesting, both to the federated learning community, as well as for designing better algorithms for LASSO, as the algorithm can be run as a parallel algorithm on $\log(d)$ threads on a single server to solve the LASSO problem."
>
> Finally, we noticed that the reviewer chose "Audience: No". Please let us know if there is anything else that we are missing.

---

### Review · Reviewer_pN2B · 2023-11-17

**Summary Of Contributions:**

The paper proposes a novel method for learning the exact support of a sparse linear regression model in a federated learning setting, where the data is distributed across multiple clients with low computational and communication capabilities. The method does not require the clients to solve any optimization problem or estimate the covariance matrix, but only uses simple calculations and one-shot communication of d bits to the centralized server. The support is estimated at the central node using majority voting and the authors provide theoretical guarantees for the exact support recovery under various conditions on the number of clients, the number of samples per client, the sparsity level, and the regularization parameter.

The method is potentially robust to client failures, model poisoning, and straggling clients, and can handle different classes of predictor variables, such as mutually independent, correlated Gaussian, or correlated sub-Gaussian. The paper also demonstrates the effectiveness of the method on synthetic and real-world datasets and shows that it can recover a similar or better support than the centralized lasso estimator without needing to aggregate all the data at the central node.

**Audience:**

Yes

**Claims And Evidence:**

Yes

**Requested Changes:**

In addition to addressing the points highlighted under weaknesses above I would also recommend illustrating some real-world applications and benefits of accurate support recovery in the introduction of the paper to illustrate the importance of the problem being solved since currently that may not be clear to a reader unfamiliar with the other literature in this place.

**Strengths And Weaknesses:**

Strengths:

1. The work proposes a novel and simple method for exact support recovery of sparse linear regression in the federated learning setup, which does not require any optimization problem or complex computation at the client level.
2. The work provides provable theoretical guarantees for the sample complexity and time complexity of the proposed method and shows that it matches or improves the optimal bounds of the centralized setting for some classes of predictor variables.

Weaknesses:

1. My main concern is that it seems from (7) and (8) that we need to know the true $w*$, $\sigma$ and $\rho$ to choose a regularization parameter $\lambda$ such that the solution satisfies the theoretical guarantees but in practice we will not know $w^{*}, \sigma, \rho$. Moreover even in the experiments $\lambda$ is set to 0.08 without any justification for how it is chosen. Please provide some explanation/heuristic that can be used to choose $\lambda$ in practice or provide an ablation study illustrating the effect of varying $\lambda$ on the solution so that we can see the consequences of choosing a "bad" $\lambda$.

2. I have a similar concern for the discussion on robustness in Section 7 where it seems like we will need to know the portion $\beta$ of clients that have gone rogue to know if we have sufficient $n_i$ to trust our output. Is it realistic to assume that we will know $\beta$ beforehand?

3. There are no experiments showing that the method is indeed robust to client failures, data poisoning or straggling even though Section 7 claims that it will be robust to these issues. I would recommend adding an experiment for at least one of these scenarios to strengthen the claims in Section 7.

4. There are also no comparisons with other federated LASSO approaches such as Smith et al. (2017b; a) from Section 1.1. Even if none of them consider support recovery explicitly, I would suggest comparing with the support of the LASSO solution to illustrate the accuracy and computational efficiency of the proposed approach over baselines.

---

> ### Author Response · Authors · 2023-11-27
>
> > W1. it seems from (7) and (8) that we need to know the true $w^*$, $\sigma$ and $\rho$ to choose a regularization parameter $\lambda$ such that the solution satisfies the theoretical guarantees... in the experiments $\lambda$ is set to 0.08... provide an ablation study illustrating the effect of varying $\lambda$...
>
> $w^*,\sigma,\rho$ are important parameters (IPs) that somehow define the problem complexity.
>
> Setting the regularization parameter $\lambda$ from IPs was done in several other works, which clearly did not prevent them from being published. To mention a few:
> - linear regression [Wainwright'09, Sharp Thresholds for High-Dimensional and Noisy Sparsity Recovery Using L1-Constrained QP]
> - learning Ising models [Ravikumar'10, High-Dimensional Ising Model Selection using L1-Regularized Logistic Regression]
> - learning Gaussian graphical models [Ravikumar'11, High-dimensional Covariance Estimation by Minimizing L1-penalized Log-determinant Divergence]
> - nonparametric regression [Ravikumar'07, Sparse Additive Models]
> - diffusion networks [Daneshmand'14, Estimating Diffusion Network Structures]
>
> As in the above papers, IPs' exact values are not needed, and only reasonable lower and upper bounds are enough to set $\lambda$. (See Section 6.)
>
> The effect of varying $\lambda$ in the real-world experiment:
> | $\lambda$ | Jaccard index Case 1 | Case 2 |
> | --- | --- | --- |
> | .04 | .46 | .56 |
> | .06 | .83 | .71 |
> | .08 | .76 | .8 |
> | .10 | .73 | .82 |
> | .12 | .76 | .75 |
> | .14 | .77 | .78 |
> | .16 | .63 | .83 |
> | .18 | .53 | .74 |
> | .2 | .42 | .76 |
>
> > W2. in Section 7... it seems like we will need to know the portion $\beta$ of clients that have gone rogue... realistic to assume that we will know $\beta$ beforehand?
>
> As argued in the previous question, the exact value of $\beta$ is not needed, and only a reasonable upper bound is enough.
>
> > W3. no experiments showing that the method is indeed robust to client failures, data poisoning or straggling... I recommend adding an experiment for at least one of these scenarios...
>
> Our next experiment shows the effect of varying $\beta$. Dimension $d=1000$, sparsity $s=3$, $g=\Omega(\log d)$ clients, $n_i=\Omega(s^2\log s)$ samples per client.
> | $\beta$ | mean exact support recovery (30 runs) |
> | --- | --- |
> | .1 | 100% |
> | .2 | 100% |
> | .3 | 100% |
> | .35 | 80% |
> | .4 | 33% |
>
> > W4. no comparisons with other federated LASSO approaches such as Smith et al. (2017b; a) from Section 1.1... I suggest comparing with the support of the LASSO solution to illustrate the accuracy and computational efficiency...
>
> We can do even better and compare to the centralized-server Lasso (Wainwright 2009b), which does not have any constraint in access to clients' data. Sparsity $s=3$, $g=\Omega(\log d)$ clients, $n_i=\Omega(s^2\log s)$ samples per client.
> | $d$ | our mean exact support recovery (30 runs) } | lasso | our mean time | lasso |
> | --- | --- | --- | --- | --- |
> | 500 | 100% | 100% | 1 | 4.1 |
> | 1000 | 100% | 100% | 2.4 | 11.6 |
> | 2000 | 100% | 100% | 4.6 | 26.7 |
>
> In addition, please see Section 6 (Choice of Regularizer) and 6.1 (An Illustrative Example) which includes, among other things, a discussion of the standard mutual incoherence condition in the centralized-server Lasso (Wainwright 2009b).
>
> > Change: I also recommend illustrating some real-world applications and benefits of accurate support recovery in the introduction... to illustrate the importance of the problem... currently that may not be clear to a reader unfamiliar with the other literature in this place
>
> For an ML practitioner, we will highlight the aspect of feature selection that is implied by support recovery.

---

### Decision · Action_Editor_kQY9 · 2023-12-18

**Recommendation:** Accept with minor revision

**Comment:**

I have selected "accept with minor revision" though the revisions I'd like to see are on the more substantial end of the spectrum.

Multiple reviewers mentioned that the manuscript generally needs to be improved for quality and ease of understanding. In particular, the organization of lemmas in the work is currently confusing. Cementing this, along with more explicit problem statements, settings of interest, and relations to prior works on LASSO (especially in distributed settings) would greatly benefit the work.

I would also ask the reviewers to clean up Section 6 in particular, on the choice of the regularizer. Multiple reviews mentioned that this is a crucial point. Therefore it seems that making this clearer would be particularly beneficial. Currently, much of Section 6 is stated informally, in a paragraph structure. I believe it'd behoove the authors to make formal theorem/lemma statements that can be pointed to directly. Additionally, better discussion of related theory in (Wainwright, 2009b) would be useful - can you extract the theorems in question and pose them in the nomenclature of this paper, to better enhance audience understanding?

**Audience:**

The reviewers were quite unanimous in this respect. The problem is interesting to people at the intersection of statistics and federated learning, and poses interesting questions that could be of interest for further exploration.

**Claims And Evidence:**

The authors generally agreed that most of the claims made in the paper were sound, and that many of those that required some clarification are reasonable to fix (eg. the issue about $\sigma^i_{jj} = 0$ for some $i$ pointed out by Reviewer y1zc). Generally, the theory seems to be sound, and the experiments complement the theory.

The one sticking point is the choice of regularization term $\lambda$. Multiple reviewers mentioned that it is (1) not clear whether the feasible region for $\lambda$ imposed by the theory is non-empty, and (2) how someone would go about choosing $\lambda$ in practice.